# Cost-effectiveness analysis of olaparib maintenance therapy for BRCA mutation ovarian cancer in the public sector in Malaysia

**Chee Meng Yong[1], Prathepamalar A. P. Yehgambaram[2], Shaun Wen Huey Lee⑩[3]***

**1** Department of Gynaecology, Hospital Ampang, Ampang, Selangor, Malaysia, **2** Radiology and Oncology Department, Hospital Kuala Lumpur, Kuala Lumpur, Malaysia, **3** School of Pharmacy, Monash University Malaysia, Subang Jaya, Selangor, Malaysia

* shaun.lee@monash.edu

**Data Availability Statement:** The underlying data used for the model prediction was obtained from unpublished data of clinical studies funded by Astra Zeneca and currently are under embargo for publication. The full data can be requested from

## Abstract

### Introduction

Ovarian cancer is one of the most common cancer among women in Malaysia. Patients with ovarian cancer are often diagnosed at an advanced stage. Despite initial response to surgery and chemotherapy, most patients will experience a relapse. Olaparib has been reported have promising effects among BRCA mutated ovarian cancer patients. This study aimed to evaluate the cost–effectiveness of olaparib as a maintenance therapy for BRCA ovarian cancer in Malaysia.

### Methods

We developed a four-state partitioned survival model which compared treatment with olaparib versus routine surveillance (RS) from a Malaysian healthcare perspective. Mature overall survival (OS) data from the SOLO-1 study were used and extrapolated using parametric models. Medication costs and healthcare resource usage costs were derived from local inputs and publications. Deterministic and probabilistic sensitivity analyses (PSA) were performed to explore uncertainties.

### Results

In Malaysia, treating patients with olaparib was found to be more costly compared to RS, with an incremental cost of RM149,858 (USD 33,213). Patients treated with olaparib increased life years by 3.05 years and increased quality adjusted life years (QALY) by 2.76 (9.45 years vs 6.40 years; 7.62 vs 4.86 QALY). This translated to an incremental cost-effectiveness ratio (ICER) of RM 49,159 (USD10,895) per life year gained and RM54,357 (USD 12,047) per QALY gained, respectively. ICERs were most sensitive to time horizon of treatment, discount rate for outcomes, cost of treatment and health state costs, but was above the RM53,770/QALY threshold.

https://vivli.org/ourmember/astrazeneca/ via ExternalDataSharing@astrazeneca.com.

**Funding:** SWHL- The study was funded by Astra Zeneca, The funders had no role in study design, data collection and analysis, decision to publish, or preparation of the manuscript.

**Competing interests:** The authors have declared that no competing interests exist.

## Conclusion

The use of olaparib is currently not a cost-effective strategy compared to routine surveillance based upon the current price in Malaysia for people with ovarian cancer with BRCA mutation, despite the improvement in overall survival.

## Introduction

Cancer is a major public health issue globally, and is a leading cause of death, accounting for nearly 10 million deaths in 2020 [1]. Importantly, in women, ovarian cancer is now the leading cause of mortality in those diagnosed with gynaecological cancer, and accounts for nearly 4% of all reported new cases diagnosed in 2020. Among the risk factors associated with ovarian cancer include increasing age, family history of breast cancer as well as smoking [2]. While early cancer detection using transvaginal ultrasound and the cancer antigen-125 (CA-125) are important strategies, they have very low predictive values [3].

In people with ovarian cancer, the current standard of care includes surgery and platinum-based chemotherapy. Nevertheless, approximately 15–20% of patients experience disease progression after completing therapy or fail to respond to therapy [4]. In addition, nearly 70% of women with advanced ovarian cancer will relapse within the next 36 months after diagnosis despite the absence of residual disease and require treatment with further courses of chemotherapy [5]. Recently, the anti-angiogenic VEGF inhibitor bevacizumab and Poly(ADP-ribose) polymerase (PARP) inhibitors have gained momentum in the management of ovarian cancer [4].

Among these, the PARP inhibitors have reported some encouraging findings in patients with ovarian cancer. Examples of PARP inhibitors include olaparib, niraparib, rucaparib as well as veliparib, of which the first three have been shown to have promising anticancer activities in BRCA mutated patients. Olaparib monotherapy was approved for first-line maintenance therapy of BRCA-mutated ovarian cancer based upon the SOLO-1 study [6]. The primary study showed that after a median follow-up of 41 months, there was a reduction in disease progression or death by 70% in the olaparib treated group (hazard ratio (HR): 0.30; 95% CI: 0.23 to 0.41). This was confirmed in the recent updated SOLO-1 study which reported that 67.0% of olaparib-treated participants were alive 7 years after randomisation compared to 46.5% among those treated with placebo [7].

Two other PARP inhibitors were subsequently approved, namely niraparib and rucaparib. In the PRIMA study, patients randomised to niraparib or placebo for 36 months or until disease progression [8]. The study found that median PFS was longer with niraparib maintenance compared to placebo (13.8 vs 8.2 months), with larger benefits among those whose tumour tested positive for HRD. The ATHENA-MONO study where patients received either rucaparib or placebo for up to 24 months or until disease progressed reported a longer PFS among those receiving rucaparib therapy (20.2 vs 9.2 months) [9]. Taken together, these results suggest that all three PARP inhibitors improved PFS and can provide benefits when used in first-line settings.

Recently updated data on SOLO-2 also found that olaparib significantly prolonged the median overall survival (OS), with a median of 51.7 months compared to 38.8 months in the placebo group [10]. Results further suggest that olaparib can provide a beneficial extension of OS in patients for platinum sensitive recurrent ovarian cancer with BRCA mutation. Despite the clinical promise of this strategy, costs are a substantial barrier to treatment accessibility

especially in low-middle income countries (LMICs) as cancer diagnosis and treatment impose a high financial burden [11,12]. Cost-effectiveness analyses have been suggested as one of the methods to assess the benefits of treatment compared to its costs. Several studies to date have examined the cost-effectiveness of introducing olaparib in patients with ovarian cancer, with varied results. For example, in a study by Shu and colleagues in China, the authors found that while olaparib led to an increase in quality adjusted life-years (QALY), the incremental cost-effectiveness (ICER) was $77,620.56/QALY, which is above what would be considered cost-effective [13]. In contrast, another similar study in Singapore, Tan *et al* found that olaparib maintenance therapy was cost-effective, with an ICER of SGD 19,822/QALY, especially among those with BRCA mutations [14].

Potential reasons for these discrepancies include the evaluations of the economic costs of cancer, which can vary across countries and settings based on local disease burden, healthcare resources availability, access to care utilization and outcomes data, as well as healthcare practices [15–17]. Currently, little is known about the economic impact of introducing olaparib maintenance therapy for women with ovarian cancer in Malaysia. This study aims to evaluate the cost-effectiveness of olaparib as a maintenance therapy following response to first-line platinum-based chemotherapy, which can be used to inform future reimbursement decision in Malaysia.

## Methods

This study was performed from January to May 2023 in accordance with the Consolidated Health Economic Evaluation Reporting Standards 2022 (CHEERS 2022) statement [18]. No ethical approval was sought since publicly available information was used.

### Decision model structure

A hypothetical four state partitioned survival with 1-month cycle, with a half-cycle correction was developed to estimate the cost-effectiveness of two management strategies, treatment with olaparib or routine surveillance in women with ovarian cancer. The mutually exclusive health states included: progression free state (PF), which can then transition to progressed disease 1 (PD1, post first progression) state, progressed disease 2 (PD2, post second progression) state and death from any cause (Fig 1). This 4-state model was chosen given that it mimicked the treatment patterns and disease progressions for advanced ovarian cancer in Malaysia and the region [14]. The model was developed on Microsoft Excel 2016 (Microsoft Corp, Redmond, WA) based upon the perspective of the Malaysian healthcare system.

In the current study, all patients entered the model in the PF state, and would receive either olaparib or routine surveillance. These patients are at risk of progression (PD1), die or could remain in the PF state. For patients who are in the PD1 state, olaparib was discontinued and patients would receive chemotherapy treatment before undergoing active surveillance and/or

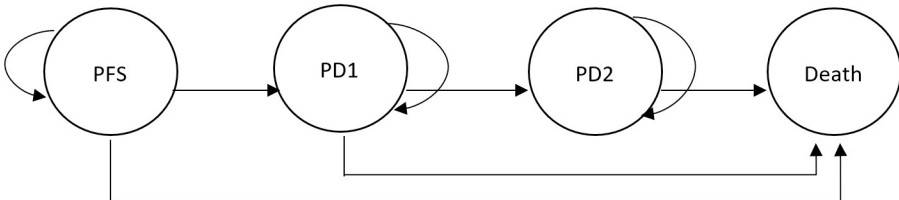

**Fig 1. Markov model structure used in the current study which simulated the four health states: Progression-free survival (PFS), post first progression (PD1), post second progression (PD2) or death.**

palliative care. Patients could subsequently stay in the PD1 state, transition into the PD2 state or die.. Patients were assumed to stay in the PD state (either PD1 or PD2) for the remaining time horizon until they died.

To obtain the health state membership in PD1 and PD2 at each time point, this was determined directly from OS curves. The PF health state was estimated from PFS curves. For instance, the proportion of patients without progression were equal to the area under the PFS1 curve. To obtain the proportion of patients post first-progression (PD1), this was calculated as the difference in the areas under the PFS2 and PFS1 curves. A similar approach was taken to obtain the proportion of patients post second-progression (PD2), where the difference in the area under the OS curve and the PFS2 curve was used (See S1 Table). These information was then used in the model to project the outcomes and costs for patients with ovarian cancer (Fig 1).

In the current study, a 28-year time horizon was considered sufficient in the base case to capture most of the survival benefits and costs accrued in the olaparib arm, given that the 10-year survival rate for patients with advanced ovarian cancer is approximately 35%, taking into account the average Malaysian women lifespan [19,20].

## Model inputs

**Clinical data.** Clinical inputs for the model were derived from SOLO-1 trial, which compared olaparib tablets with placebo after completing at platinum-based chemotherapy regimen [6]. Due to the lack of patient-level data, the PFS estimates were based upon the Kaplan-Meier (KM) estimates for the first 24 months. However, diagnostic plots indicated that the proportional hazards assumption was violated for all other timepoints beyond 24 months. To address this, we adopted an independent 'piecewise' model approach, where standard parametric survival distributions (including exponential, Weibull, log-logistic, log-normal, Gompertz, and generalized gamma) were used to fit onto each Kaplan-Meier curve beyond the 24-month inflexion point. Based upon the goodness-of-fit statistics (Akaike Information Criterion, AIC) as well as visual inspection of the curves, and clinical opinion, the most appropriate distributions for the extrapolation was selected. Similar to PFS, the OS estimates for olaparib during the first 24 months were based on the KM estimates from SOLO-1 study. As such, we used a similar approach as described above to determine the OS estimates beyond 24 months (S1 Table).

**Adverse events.** The current model only considered the disutility and cost impact related to those of grade 3 to 5 adverse events (AEs) since it was associated with significant costs, requires longer hospitalisation and impacts quality of life. Specifically, only AEs with a frequency of ≥5% from SOLO1 were included in the model (Table 1).

**Health state utilities.** Due to the absence of locally available data, the utility values for PFS and PD1 states were derived from SOLO-1 study [6], while PD2 were obtained from utility data of patients that progressed after treatment with olaparib in a platinum sensitive recurrent ovarian cancer [26]. Any reduction in utility due to AEs was calculated as a weighted sum of the utility associated with the included AEs. The weights were the reported frequencies of the AEs in the corresponding arm in SOLO-1.

## Resource utilization and costs

All costs were considered from a Malaysian health care perspective, and only direct costs were considered. Costs included in the model are those associated with drug acquisition, administration, monitoring, and adverse effect based upon the Malaysian Government Scheduled fees [21]. All drug costs included in the model reflects the discount and any rebates given, based

**Table 1. Modelled characteristics and costs imputed into the model.**

| Parameter | Value | Reference |
|---|---|---|
| Age, years (standard error) | 54.2 (5.4) | Local clinical expert opinion |
| Direct medical cost, RM (USD) | | |
| Consultation (office visit) | 100.00 (22.16) | Malaysian Government Scheduled fees [21] |
| Blood count | 40.00 (8.87) | |
| Liver function test | 100.00 (22.16) | |
| Renal profile | 80.00 (17.73) | |
| CA125 tumour marker | 75.00 (16.62) | |
| Chest computer tomography | 675.00 (149.60) | |
| Olaparib (per month) | 13818.63 (3062.50) | IMS data |
| Subsequent-line platinum chemotherapy (per cycle) | 714.33 (158.32) | IMS data |
| Subsequent-line non- platinum chemotherapy (per cycle) | 14027.76 (3108.99) | IMS data |
| Intravenous drug administration (per administration) | 400.00 (88.65) | Malaysian Government Scheduled fees [21] |
| BRCA1/2 test (per patient) | 2000.00 (443.26) | Local clinical expert |
| Anaemia management | 4700.11 (1041.69) | Azmi 2018 [22] |
| Neutropenia management | 3203.04 (709.89) | Ariffin 2001[23] |
| Diarrhoea management | 258.85 (57.37) | Loganathan 2015 [24] |
| Hospice care (per event) | 2000.00 (443.26) | Local clinical expert |
| Health state utility | | |
| Progression free | 0.82 | SOLO1 trial [25] |
| 1st progressive disease | 0.77 | SOLO1 trial [25] |
| 2nd progressive disease | 0.68 | NICE TA381 [26] |

upon the negotiated prices for the public sector. The healthcare resources include office visits, blood count, and computed tomography (CT) scans, based upon clinical guidelines and clinical input. The model assumes different resource use rates for all stages, reflecting clinical practice. Similarly, the cost of testing for BRCA was accounted for as a one-off cost for all patients receiving olaparib, assuming a 13.9% BRCA positive rate [27].

A dose of 600mg/day olaparib was used in the model based upon data from SOLO-1, with a maximum duration of 2 years before olaparib was stopped. Since olaparib was orally administered, no administration cost was considered in the model. The proportion of patients receiving olaparib over time was determined based upon the PFS curve for time to discontinuation or death in SOLO-1 study. Women who have disease progressed will then receive subsequent treatment, which includes a mixture of platinum and non-platinum chemotherapy regimens, which was identified through local clinical expert survey. For each chemotherapy cycle, the cost of consultation, blood counts and administration cost were accounted for. All patients with chemotherapy was assumed to receive 6 cycles of treatment (S2 Table). The utilization of each chemotherapy regimen can be found in S3 Table.

The proportion of women transitioning from first-line maintenance therapy to second-line subsequent therapy was obtained from SOLO-1. For women who transition to death, they were assumed to incur a one-time cost associated with palliative care, and assumed to be the same across arms [28]. All costs were adjusted to 2022 prices based upon the consumer price index [29].

## Outcomes

The outcomes of interest were costs, overall life years (LYs), progression-free life years (PFLYs), quality-adjusted life years (QALYs), and the incremental cost-effectiveness ratio (ICER). All costs were discounted at 3% per annum.

## Sensitivity analyses

We conducted 1-way deterministic sensitivity analyses on key model parameters to investigate the impact of each parameter and to account for the uncertainties. The key parameters varied are listed in S4 Table. Whenever possible, we used the standard error of the selected parameter based upon the data source used to inform the mean values. In the event these are unavailable, we varied them by 20% of the mean values. These parameters were tested individually to determine the impact based upon the recommended willingness to pay (WTP) threshold of one-GDP of Malaysia in 2022 (RM53,043/QALY or USD 11,756; assuming 1USD = RM4.512) [30,31]. A probabilistic sensitivity analysis (PSA) was also conducted using 1,000 Monte Carlo simulation to assess the uncertainty in model input with the base-case results.

# Results

## Base case analysis

Over a 28-year time horizon, treating women with ovarian cancner wih olaparib was more costly compared to routine surveillance. Women receiving olaparib incurred an additional incremental cost of RM 149,858 (USD 33,213) compared to routine surveillance. This was mainly due to drug acquisition cost of olaparib, accounting for the majority of incremental costs (RM 280,673; USD 62,206), but this costs was offset by savings from subsequent management costs (RM 147,234; USD 32,632). Patients on olaparib lived 9.45 years compared to 6.40 years for those with routine surveillance, as they had a longer progression-free life years of 5.00 and 4.10 progression-free QALY. This led to greater life-year of 3.05 and incremental QALY of 2.76 compared to routine surveillance, which translated to an ICER of RM 49,159 (USD 10,895) per LY gained and RM 54,357 (USD 12,047) per QALY gained (Table 2).

## Sensitivity analyses

One-way sensitivity analyses showed that ICER values were most sensitive to the time horizon of treatment, discount rate for outcomes, cost of treatment as well as health state costs (Fig 2). ICER values ranged between RM 35,578 (USD 7,885) to RM76,007 (USD 16,846) when parameters were varied over the range of possible values assumed. The range remained above the RM 53,770 per QALY gained for olaparib compared to routine surveillance in most instances, except when the cost of olaparib was reduced, or when disease progressed slower in those treated with olaparib. The results of the ICER values when various parameters were varied are shown in S5 Table.

Probabilistic sensitivity analyses with 1000 simulated patients found that olaparib remained congruent with the deterministic analysis results, which found that olaparib was a more costly strategy. The use of olaparib was estimated to increase quality-adjusted life years by 2.62 (95% credible interval (CrI): 0.77 to 4.18) and life-years by 2.87 (0.35 to 4.96). The mean probabilistic ICER was RM 50,054 per LY gained and RM 54,815 (USD 12,149) per QALY gained. Scatterplot showed that over 99% of the iterations were within the north-east quadrant, suggesting that olaparib use was more effective but more costly than usual care. In particular, at a willingness-to-pay threshold of RM53,043 olaparib had a 46% chance of being cost effective versus routine surveillance (Fig 3).

**Table 2. Results of the cost-effectiveness study.**

|  | Olaparib | Standard of care | Difference |
|---|---|---|---|
| **Effectiveness** |  |  |  |
| **Total LYS** | 9.45 | 6.41 | 3.05 |
| Progression free survival | 8.08 | 3.08 | 5.00 |
| Period until first progression | 0.55 | 1.70 | -1.14 |
| Period until second progression | 0.82 | 1.63 | -0.81 |
| **Total QALYs** | 7.62 | 4.86 | 2.76 |
| Progression free survival | 6.63 | 2.53 | 4.10 |
| Period until first progression | 0.43 | 1.22 | -0.79 |
| Period until second progression | 0.56 | 1.11 | -0.55 |
| **Cost, RM (USD)** |  |  |  |
| Total cost of first-line maintenance treatment and subsequent therapies | 326,638 (72,393) | 193,200 (42,819) | 133,439 (29,574) |
| Cost of PFS state (first 2 years) | 5,234 (1,160) | 2,156 (478) | 3,078 (682) |
| Cost of PFS state (subsequent years) | 4,159 (922) | 1,529 (339) | 2,630 (583) |
| Cost of PD1 state | 1,348 (299) | 3,862 (856) | - 2,514 (-557) |
| Cost of PD2 state | 2,000 (443) | 3,981 (882) | - 1,981 (-439) |
| Cost of terminal disease | 649 (144) | 833 (185) | - 184 (-41) |
| Cost of adverse events | 1,162 (258) | 171 (38) | 992 (220) |
| Cost of *BRCA* testing | 14,400 (3191) | 0 (0) | 14,400 (3191) |
| **ICERS** |  |  |  |
| Incremental cost per QALY gained (RM/QALY) |  |  | 54,357 (12,047) |
| Incremental cost per LY gained (RM/LY) |  |  | 49,159 (10,895) |

Assumes 1USD = RM4.512.

OS: Overall survival; PF: Progression free; PD1:, Post first progression; PD2:, Post second progression; ICER: Incremental cost-effectiveness ratio.

## Discussion

The use of PARPis have been recently shown to be effective in delaying the progression of ovarian cancer. Among them, olaparib could prolonged PFS of those with BRCA mutant ovarian cancer patients. Given this clinical promise, we undertook a cost-effectiveness analysis to

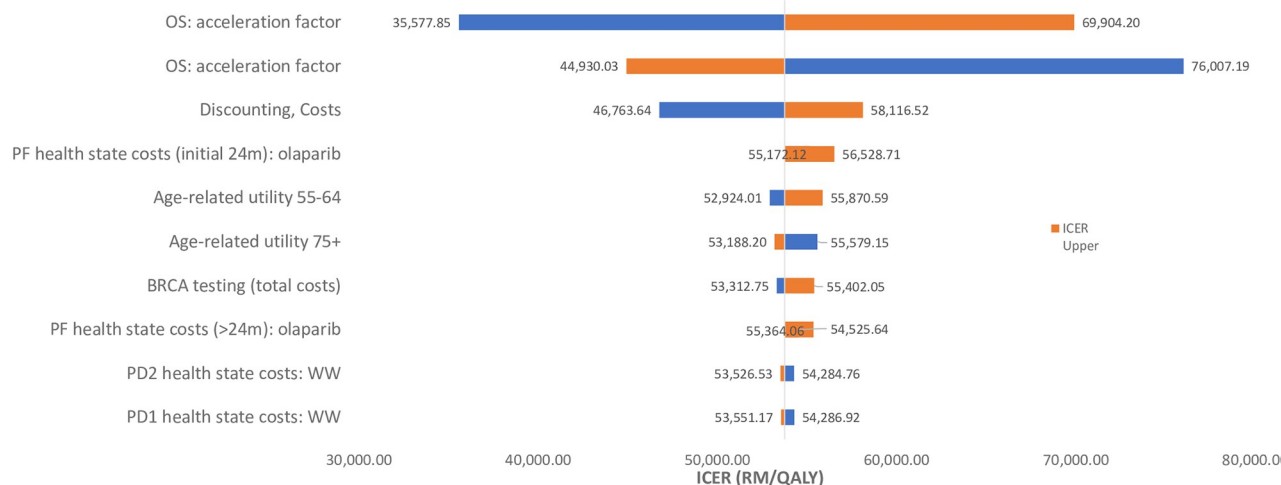

**Fig 2. One-way deterministic sensitivity analysis showing the 10 parameters which the model was most sensitive.**

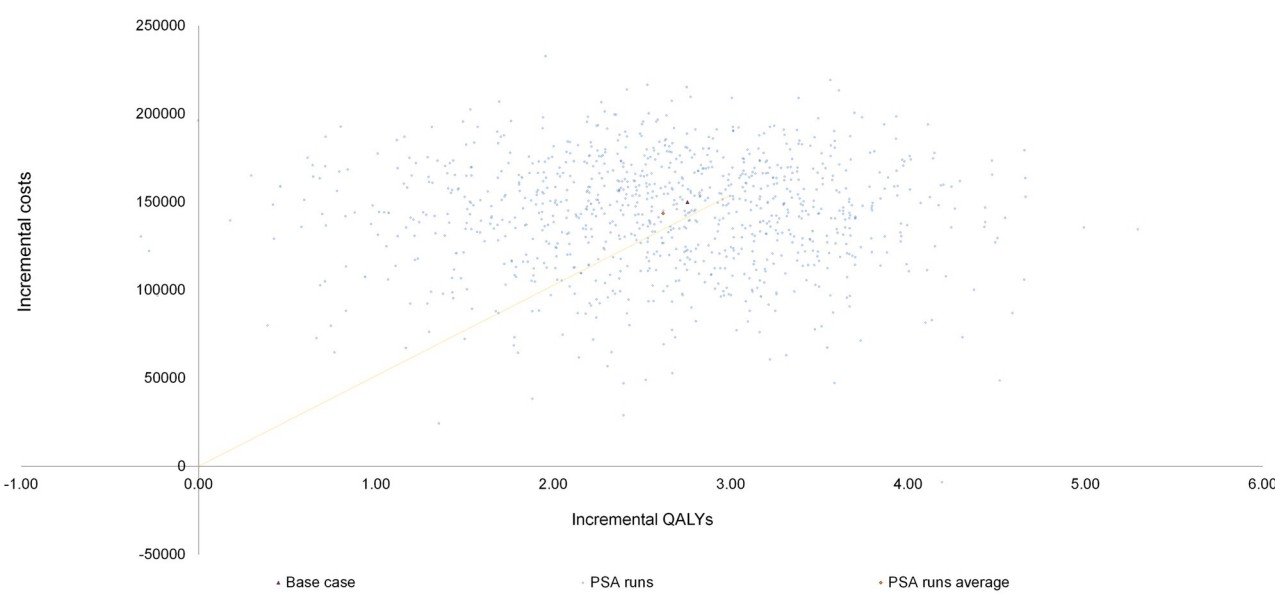

**Fig 3. Probabilistic sensitivity analysis: Cost effectiveness plane.**

evaluate the economic impact of using olaparib. In our cost-effectiveness analysis of olaparib compared to routine surveillance in Malaysia, treatment with olaparib substantially improves the overall PFS by 5.00 years, life-years by 3.05 years and quality-adjusted life years by 2.78. These results are in-line with the recent results reported in the SOLO1 trial, which showed that maintenance therapy with olaparib can improve the long-term overall survival of patients. Importantly, despite treatment with olaparib capped at 2 years, many patients continue to experience benefits beyond this point [10,25].

However, our analyses showed that olaparib was not likely to be a cost-effective strategy for women with ovarian cancer at the current price from Malaysia's context, as it was above the Malaysian threshold of cost-effectiveness, which suggested that the value should be below one GDP. Results of the one-way sensitivity analysis showed that the ICER was highly sensitive to the cost of olaparib. Overall, the findings suggest that a reduction in the cost of olaparib is needed to be a cost-effective treatment in Malaysia. Nevertheless, this needs to be taken into context the substantial improvements in QALY and LYs relative to current practice. Indeed, as shown in the SOLO1 trial, treatment with olaparib was shown to provide significant benefits beyond the 2 years treatment period. Results of our study closely mimic the study by Muston and colleagues in the United States which found that the use of olaparib compared to routine surveillance in US increases both the LYs and QALYs of women with newly diagnosed advanced ovarian cancer and with a germline or somatic BRCA mutation [32].

This study offers several strengths. This is the first economic analysis conducted to evaluate the cost-effectiveness of olaparib maintenance therapy versus placebo in ovarian cancer patients with BRCA mutations in Malaysia. Our study showed that the use of olaparib increased cost by RM150,197 compared to routine surveillance, which is above the proposed willingness-to-ratio of RM47,600 in Malaysia. Another strength of our study is that the study was informed by the latest results from the SOLO1 study [25], which further lends credibility to our model. In addition, our model uses locally derived data such as the life-table whenever possible.

Nevertheless, there are several limitations to our study. As with most studies, we extrapolated the survival data beyond the follow-up duration of the trial, which may not accurately reflect the real-world condition. However, feedback from our key experts suggest that the survival model mimics the real-world experience at their setting, which provides some validation of our results. Nevertheless, the model uncertainty rates were relatively small based on the goodness of fit model. In addition, our model did not evaluate the impact of using different chemotherapies used, as the effects were unavailable. Like most other cost-effectiveness analysis, we had made several assumptions based upon the SOLO1 study which may introduce inconsistencies between treatment cost and their effects. Finally, our model only considered cost related to grade 3 or 4 adverse events, which were more likely to be costly. Finally, we used utility scores of those derived from the SOLO1 study and literature review as these values were unavailable for Malaysia.

## Conclusion

The use of olaparib as a maintenance therapy is currently not a cost-effective strategy compared to standard of care which is routine surveillance in Malaysia, especially among those with BRCA mutation using a threshold of RM53,043/QALY. However, given that olaparib improves QALY and LYs, it may be important to reevaluate the cost-effectiveness of olaparib in the future

## Supporting information

**S1 Checklist. CHEERS 2022 checklist.**
(DOCX)

**S1 Table. The rate of transition used in the model.**
(DOCX)

**S2 Table. List of subsequent line anticancer regimens used in the model.**
(DOCX)

**S3 Table. Resource utilization for subsequent therapies.**
(DOCX)

**S4 Table. Parameters included in the deterministic and probabilistic analysis and their distribution.**
(DOCX)

**S5 Table. Results of deterministic sensitivity analyses conducted in this study.**
(DOCX)

## Author Contributions

**Conceptualization:** Chee Meng Yong, Shaun Wen Huey Lee.

**Data curation:** Prathepamalar A. P. Yehgambaram, Shaun Wen Huey Lee.

**Formal analysis:** Chee Meng Yong, Prathepamalar A. P. Yehgambaram, Shaun Wen Huey Lee.

**Funding acquisition:** Shaun Wen Huey Lee.

**Methodology:** Chee Meng Yong.

**Project administration:** Shaun Wen Huey Lee.

**Software:** Shaun Wen Huey Lee.

**Supervision:** Shaun Wen Huey Lee.

**Validation:** Prathepamalar A. P. Yehgambaram, Shaun Wen Huey Lee.

**Writing – original draft:** Shaun Wen Huey Lee.

**Writing – review & editing:** Chee Meng Yong, Prathepamalar A. P. Yehgambaram, Shaun Wen Huey Lee.

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
