## [Decision Letter · Decision Letter 0]

13 Nov 2023

PONE-D-23-29905Cost-effectiveness analysis of olaparib maintenance therapy for BRCA mutation ovarian cancer in the public sector in MalaysiaPLOS ONE

Dear Dr. Lee,

Thank you for submitting your manuscript to PLOS ONE. After careful consideration, we feel that it has merit but does not fully meet PLOS ONE’s publication criteria as it currently stands. Therefore, we invite you to submit a revised version of the manuscript that addresses the points raised during the review process.

We look forward to receiving your revised manuscript.

Kind regards,

Alvaro Galli

Academic Editor

PLOS ONE

https://www.tandfonline.com/doi/abs/10.1080/14737167.2021.1890587

In your revision ensure you cite all your sources (including your own works), and quote or rephrase any duplicated text outside the methods section. Further consideration is dependent on these concerns being addressed.

Reviewers' comments:

Reviewer's Responses to Questions

**Comments to the Author**

1. Is the manuscript technically sound, and do the data support the conclusions?

Reviewer #1: Yes

2. Has the statistical analysis been performed appropriately and rigorously? 

Reviewer #1: I Don't Know

3. Have the authors made all data underlying the findings in their manuscript fully available?

Reviewer #1: No

4. Is the manuscript presented in an intelligible fashion and written in standard English?

Reviewer #1: No

5. Review Comments to the Author

Reviewer #1: Introduction

- The authors indicate that a number of PARPi's are available as first-line maintenance therapies. Why focus here on only olaparib?

- First sentence of para 4 is unclearly written. The relevance of data for olaparib in PSROC, a setting later than the one considered in this paper is unclear.

- Sentence 2 of para 5 requires a supporting reference. Suggested where?

Methods

- The rationale for this model structure is unclear. In this evaluation of olaparib, the authors cite structures from HTA evaluations of niraparib. This seems potentially disingenuous if HTA evaluations of olaparib, the subject of this paper, used a different structure. Certainly, Muston et al used 3 states, not 4 states as here [32]. Why use 4 states if others have used 3?

- Paragraph 3 explaining health state membership is rather unclear, compounded by Suppl Table 1. We are told this is a 4 state partitioned survival model with states of PF, PD1, PD2 and death. This structure requires survival modeling of PFS1, PFS2 and OS. The supplementary table provides other information that does not correspond with this. The parameter values should be reported. The textual description in the manuscript is rather unclear.

- 28 seems a curious number to pick for the time horizon (years). Why not 20, 25, 30 or 40? The time horizon may be missing from the OWSA.

- Health state utilities. Please confirm the references - some are included in Table 1, but not the text. What value sets / country do these relate to? Two sources are used, contrary to ISPOR guidelines because they may not be consistent. Why were niraparib utilities used?

- Table 1. What kind of drug cost is quoted in Table 1 for olaparib. 'IMS' is indicated as a reference, but no full reference is actually provided. Is this derived from a public list price? Or does this value already reflect discounts/rebates that may be available and be a net price? This is important to know in understanding the conclusion from this paper.

- Resource utilization information is missing from the section on resource utilization. The section describes what resources are assumed, but not how much of each. For example, what subsequent treatments were recommended in the clinician survey? How frequently are physician visits required?

- Sensitivity analyses. Why were only 'key' parameters varied? How was it pre-determined which parameters were 'key'? What is the difference between a PFS model and probability in this context? Was OS not varied? What statistical distributions were used for the PSA?

Results

- Sentence 2 is unclear as it contains two instances of 'which'.

- Sensitivity analyses, paragraph 2. Articles 'a' and 'the' are occasionally missing.

- What is the relevance of a WTP threshold of RM53,043? Is this a local standard? Please explain. Otherwise it is little surprise that about half of PSA simulations were above/below a value so close to the mean ICER.

- Total LYs and total QALYs should be added to Table 2.

Discussion

- What is a 'quality of life year' (line 6)? Is it related to QALY?

- Life years appear to increase by either 3.05 years (line 6) or 2.87 years (line 8). Which?

- Para 2 cites a slightly confusing mix of studies. The most relevant study here by treatment setting appears to be Muston [32], since this also evaluates olaparib in this same treatment setting (but in the US). It is questionable whether the others mentioned are truly 'similar studies' as claimed [10, 29-31] because they mostly appear in later PROC settings. One cannot easily compare CE of an agent between two different treatment settings - or the substantial difficulties with this should be acknowledged.

- The assumption of ToT=PFS (but capped at 2 years) may also be key to the results and should ideally be explored. What if ToT was longer/shorter than PFS?

Conclusion

- Not well worded. This suggests patients ('those') have a 53,043 (no units) /QALY threshold. 'The need' in sentence 2 is unclear.

Supplementary information

- Table 1 is poorly designed and unhelpful. See also comment above. The authors should disclose the statistical distributions, parameters, and ideally variance matrix for the three survival curves required in this model - PFS1, PFS2 and OS. Since two piece modeling is used, PFS1, PFS2 and OS survival at the 24 month cutoff (per Methods model input) is also important to disclose.

6. PLOS authors have the option to publish the peer review history of their article (what does this mean?). If published, this will include your full peer review and any attached files.

Reviewer #1: No

---

## [Author Response · Author response to Decision Letter 0]

25 Dec 2023

18 December 2023

Alvaro Galli

Academic Editor

PLOS ONE

Dear Prof Galli

Thank you very much for the opportunity to have our publication submitted and reviewed by PLOS One. As requested by the reviewers, we are pleased to submit our revised article entitled “Cost-effectiveness analysis of olaparib maintenance therapy for BRCA mutation ovarian cancer in the public sector in Malaysia” for review by PLOS One. As outlined below, we have made the following modifications in our article as follow:-

Reviewer #1 had the following comments

Comment#1 The authors indicate that a number of PARPi's are available as first-line maintenance therapies. Why focus here on only olaparib?

We take note and thank the reviewer for the suggestion. We have now expanded our introduction section to describe the other PARPi that are currently available. Our revised introduction now reads as follow:

Among these, the PARP inhibitors have reported some encouraging findings in patients with ovarian cancer. Examples of PARP inhibitors include olaparib, niraparib, rucaparib as well as veliparib, of which the first three have been shown to have promising anticancer activities in BRCA mutated patients. Olaparib monotherapy was approved for first-line maintenance therapy of BRCA-mutated ovarian cancer based upon the SOLO-1 study.[6] The primary study showed that after a median follow-up of 41 months, there was a reduction in disease progression or death by 70% in the olaparib treated group (hazard ratio (HR): 0.30; 95% CI: 0.23 to 0.41). This was confirmed in the recent updated SOLO-1 study which reported that 67.0% of olaparib-treated participants were alive 7 years after randomisation compared to 46.5% among those treated with placebo.[7] 

Two other PARP inhibitors were subsequently approved, namely niraparib and rucaparib. In the PRIMA study, patients randomised to niraparib or placebo for 36 months or until disease progression The study found that median PFS was longer with niraparib maintenance compared to placebo (13.8 vs 8.2 months), with larger benefits among those whose tumour tested positive for HRD. The ATHENA-MONO study where patients received either rucaparib or placebo for up to 24 months or until disease progressed reported a longer PFS among those receiving rucaparib therapy (20.2 vs 9.2 months). Taken together, these results suggest that all three PARP inhibitors improved PFS and can provide benefits when used in first-line settings.

Comment #2 First sentence of para 4 is unclearly written. The relevance of data for olaparib in PSROC, a setting later than the one considered in this paper is unclear.

We agree and thank the reviewer for the suggestion. We have now deleted that sentence to avoid any confusion.

Comment # 3. Sentence 2 of para 5 requires a supporting reference. Suggested where?

We agree this was unclear and thank the reviewer for the suggestion. We have now included 2 references to clarify this. The revised sentence and references are as follow

Despite the clinical promise of this strategy, costs are a substantial barrier to treatment accessibility especially in low-middle income countries (LMICs) as cancer diagnosis and treatment impose a high financial burden.[1, 2]

Reference

• Aminuddin F, Bahari MS, Zainuddin NA, et al. The Direct and Indirect Costs of Cancer among the Lower-Income Group: Estimates from a Pilot and Feasibility Study. Asian Pac J Cancer Prev 2023;24(2):489-96. doi: 10.31557/apjcp.2023.24.2.489

• Leive A, Xu K. Coping with out-of-pocket health payments: empirical evidence from 15 African countries. Bull World Health Organ 2008;86(11):849-56. doi: 10.2471/blt.07.049403

Comment #4. The rationale for this model structure is unclear. In this evaluation of olaparib, the authors cite structures from HTA evaluations of niraparib. This seems potentially disingenuous if HTA evaluations of olaparib, the subject of this paper, used a different structure. Certainly, Muston et al used 3 states, not 4 states as here [32]. Why use 4 states if others have used 3?

We agree and thank the reviewer for pointing this out. We have now revised our references to cite the relevant HTA evaluations for Olaparib instead. We also do note that our study used a 4 state as opposed to a 3-state model used by other HTA. However, our consultation with clinicians suggest that a 3-state model was considered oversimplistic and does not reflect the disease progression of ovarian cancer, which led to our use of a 4-state model. This similar approach has been used by Tan and colleagues in Singapore whose healthcare system mimics those in Malaysia. Our revised methods now reads as follow:-

A hypothetical four state partitioned survival with 1-month cycle, with a half-cycle correction was developed to estimate the cost-effectiveness of two management strategies, treatment with olaparib or routine surveillance in women with ovarian cancer. The mutually exclusive health states included: progression free state (PF), which can then transition to progressed disease 1 (PD1, post first progression) state, progressed disease 2 (PD2, post second progression) state and death from any cause (Figure 1). This 4-state model was chosen given that it mimicked the treatment patterns and disease progressions for advanced ovarian cancer in Malaysia and the region.[3]

Additional details are also included as part of clinical data inputs as follow:-

Clinical inputs for the model were derived from SOLO-1 trial, which compared olaparib tablets with placebo after completing at platinum-based chemotherapy regimen.[4] Due to the lack of patient-level data, the PFS estimates were based upon the Kaplan-Meier (KM) estimates for the first 24 months. However, diagnostic plots indicated that the proportional hazards assumption was violated for all other timepoints beyond 24 months. To address this, we adopted an independent ‘piecewise’ model approach, where standard parametric survival distributions (including exponential, Weibull, log-logistic, log-normal, Gompertz, and generalized gamma) were used to fit onto each Kaplan-Meier curve beyond the 24-month inflexion point. Based upon the goodness-of-fit statistics (Akaike Information Criterion, AIC) as well as visual inspection of the curves, and clinical opinion, the most appropriate distributions for the extrapolation was selected. Similar to PFS, the OS estimates for olaparib during the first 24 months were based on the KM estimates from SOLO-1 study. As such, we used a similar approach as described above to determine the OS estimates beyond 24 months (Supplementary Table 1). 

Comment #5 Paragraph 3 explaining health state membership is rather unclear, compounded by Suppl Table 1. We are told this is a 4 state partitioned survival model with states of PF, PD1, PD2 and death. This structure requires survival modeling of PFS1, PFS2 and OS. The supplementary table provides other information that does not correspond with this. The parameter values should be reported. The textual description in the manuscript is rather unclear.

We apologise for the confusion. We have now included the model parameters we had used in our revised appendix to provide a better context and transparency. We hope this will address the reviewer’s concern

Comment #6 28 seems a curious number to pick for the time horizon (years). Why not 20, 25, 30 or 40? The time horizon may be missing from the OWSA.

We agree this was unclear. In our model, we had chosen 28 years since the 10-year survival rates for ovarian cancer were 35%, and most women were not expected to live beyond 80 years old. As such, we had used this as our maximum duration. We have now included the rationale in our methods as follow:-

In the current study, a 28-year time horizon was considered sufficient in the base case to capture most of the survival benefits and costs accrued in the olaparib arm, given that the 10-year survival rate for patients with advanced ovarian cancer is approximately 35%, taking into account the average Malaysian women lifespan.[5, 6]

We have also updated the figure and included the time horizon in our OWSA

Comment #7 Health state utilities. Please confirm the references - some are included in Table 1, but not the text. What value sets / country do these relate to? Two sources are used, contrary to ISPOR guidelines because they may not be consistent. Why were niraparib utilities used?

We do note on the reviewer’s concern. In the absence of data from SOLO1, the quality of life of patients with PD2 was based on utility data from patients that progressed after treatment with olaparib in a platinum sensitive recurrent OC setting, as reported in NICE TA381. These data represent the quality of life of patients with that have progressed after 2nd line PARP treatment. We do note that these reference was incorrect and have now revised it in our revised table.

The revised sentence now reads as follow:-

Due to the absence of locally available data, the utility values for PFS and PD1 states were derived from SOLO-1 study,[4] while PD2 were obtained from utility data of patients that progressed after treatment with olaparib in a platinum sensitive recurrent ovarian cancer[7]

Comment #8 Table 1. What kind of drug cost is quoted in Table 1 for olaparib. 'IMS' is indicated as a reference, but no full reference is actually provided. Is this derived from a public list price? Or does this value already reflect discounts/rebates that may be available and be a net price? This is important to know in understanding the conclusion from this paper.

We agree this was unclear. The cost included in the current model are those that already reflects the discount and rebates given and are the purchasing price for the public sector in Malaysia. We have now clarified this in our revised manuscript as follow:-

All drug costs included in the model reflects the discount and any rebates given, based upon the negotiated prices for the public sector

Comment #9- Resource utilization information is missing from the section on resource utilization. The section describes what resources are assumed, but not how much of each. For example, what subsequent treatments were recommended in the clinician survey? How frequently are physician visits required?

We agree this was unclear and thank the reviewer for pointing this out. We have now included this information as follow and as part of Supplementary Table 3-

A dose of 600mg/day olaparib was used in the model based upon data from SOLO-1, with a maximum duration of 2 years before olaparib was stopped. Since olaparib was orally administered, no administration cost was considered in the model. The proportion of patients receiving olaparib over time was determined based upon the PFS curve for time to discontinuation or death in SOLO-1 study. Women who have disease progressed will then receive subsequent treatment, which includes a mixture of platinum and non-platinum chemotherapy regimens, which was identified through local clinical expert survey. For each chemotherapy cycle, the cost of consultation, blood counts and administration cost were accounted for. All patients with chemotherapy was assumed to receive 6 cycles of treatment (Supplementary Table 2). The utilization of each chemotherapy regimen can be found in Supplementary Table 3.

Comment #10. Sensitivity analyses. Why were only 'key' parameters varied? How was it pre-determined which parameters were 'key'? What is the difference between a PFS model and probability in this context? Was OS not varied? What statistical distributions were used for the PSA?

We agree this was unclear. We have now included the list of the various parameters varied as part of our deterministic analysis as well as the parameters chosen for PSA and their distribution in Supplementary Tabe 4 to increase the reproducibility and transparency of our model. The effects on changes in overall survival was also examined. We apologise for the lack of clarity. 

Comment #11. - Sentence 2 is unclear as it contains two instances of 'which'.

We agree. We have now revised the sentence to read as follow:-

Over a 28-year time horizon, treating women with ovarian cancner wih olaparib was more costly compared to routine surveillance. Women receiving olaparib incurred an additional incremental cost of RM 149,858 (USD 33,213) compared to routine surveillance. This was mainly due to drug acquisition cost of olaparib, accounting for the majority of incremental costs (RM 280,673; USD 62,206) , but this costs was offset by savings from subsequent management costs (RM 147,234; USD 32,632)

Comment #12. Sensitivity analyses, paragraph 2. Articles 'a' and 'the' are occasionally missing.

Thank you for pointing this out. We have revised our sentence structure of our results section to clarify this. The revised paragraph now reads as follow:-

We conducted 1-way deterministic sensitivity analyses on key model parameters to investigate the impact of each parameter and to account for the uncertainties. The key parameters varied are listed in Supplementary Table 4. Whenever possible, we used the standard error of the selected parameter based upon the data source used to inform the mean values. In the event these are unavailable, we varied them by 20% of the mean values. These parameters were tested individually to determine the impact based upon the recommended willingness to pay (WTP) threshold of one-GDP of Malaysia in 2022 (RM53,043/QALY or USD 11,756; assuming 1USD=RM4.512).[30, 31] A probabilistic sensitivity analysis (PSA) was also conducted using 1,000 Monte Carlo simulation to assess the uncertainty in model input with the base-case results.

Comment #13. What is the relevance of a WTP threshold of RM53,043? Is this a local standard? Please explain. Otherwise it is little surprise that about half of PSA simulations were above/below a value so close to the mean ICER.

We apologise that this was unclear. In Malaysia, the WTP threshold was set as one GDP by the Malaysian Health Technology Assessment Workgroup and has been used as a local standard for health technology assessment. We have now clarified this in the methods to avoid any ambiguity. The revised sentence now reads as follow

These parameters were tested individually to determine the impact based upon the recommended willingness to pay (WTP) threshold of one-GDP of Malaysia in 2022 (RM53,043/QALY or USD 11,756; assuming 1USD=RM4.512).[8, 9]

Comment #14. Total LYs and total QALYs should be added to Table 2.

We thank the reviewer for pointing this out as it was unclear. We have now revised our heading on Table 2 to clarify this 

Comment #15. What is a 'quality of life year' (line 6)? Is it related to QALY?

We thank the reviewer for pointing this out. We have now revised our sentence to describe this as quality-adjusted life year as intended. The revised sentence now reads as follow:-

In our cost-effectiveness analysis of olaparib compared to routine surveillance in Malaysia, treatment with olaparib substantially improves the overall PFS by 5.00 years, life-years by 3.05 years and quality-adjusted life years by 2.78

Comment #16. Life years appear to increase by either 3.05 years (line 6) or 2.87 years (line 8). Which?

We apologise for the confusion. We had initially wanted to illustrate the similarities of our PSA results with the base case but we do realise that this had resulted in further confusion. To avoid this confusion, we have now deleted the sentence.

Comment #17. Para 2 cites a slightly confusing mix of studies. The most relevant study here by treatment setting appears to be Muston [32], since this also evaluates olaparib in this same treatment setting (but in the US). It is questionable whether the others mentioned are truly 'similar studies' as claimed [10, 29-31] because they mostly appear in later PROC settings. One cannot easily compare CE of an agent between two different treatment settings - or the substantial difficulties with this should be acknowledged.

We take note of the reviewer’s suggestion to improve the clarity of our manuscript. We have now revised our paragraph 2 and removed the studies that are not considered similar or have a different setting. We have also included the limitations as the reviewer suggested. 

However, our analyses showed that olaparib was not likely to be a cost-effective strategy for women with ovarian cancer at the current price from Malaysia’s context, as it was above the Malaysian threshold of cost-effectiveness, which suggested that the value should be below one GDP. Results of the one-way sensitivity analysis showed that the ICER was highly sensitive to the cost of olaparib. Overall, the findings suggest that a reduction in the cost of olaparib is needed to be a cost-effective treatment in Malaysia. Nevertheless, this needs to be taken into context the substantial improvements in QALY and LYs relative to current practice. Indeed, as shown in the SOLO1 trial, treatment with olaparib was shown to provide significant benefits beyond the 2 years treatment period. Results of our study closely mimic the study by Muston and colleagues in the United States which found that the use of olaparib compared to routine surveillance in US increases both the LYs and QALYs of women with newly diagnosed advanced ovarian cancer and with a germline or somatic BRCA mutation.[32]

Comment #18. The assumption of ToT=PFS (but capped at 2 years) may also be key to the results and should ideally be explored. What if ToT was longer/shorter than PFS?

We do agree this is an interesting point. However, as most of the information was unavailable, introducing this variable may result in more uncertainties in our model. 

Comment #19. Not well worded. This suggests patients ('those') have a 53,043 (no units) /QALY threshold. 'The need' in sentence 2 is unclear.

We thank the reviewer for the suggestion. We have now revised our conclusion to read as follow

The use of olaparib as a maintenance therapy is currently not a cost-effective strategy compared to standard of care which is routine surveillance in Malaysia, especially among those with BRCA mutation using a threshold of RM53,043/QALY. However, given that olaparib improves QALY and LYs, it may be important to reevaluate the cost-effectiveness of olaparib in the future 

Comment #20. Supplementary information - Table 1 is poorly designed and unhelpful. See also comment above. The authors should disclose the statistical distributions, parameters, and ideally variance matrix for the three survival curves required in this model - PFS1, PFS2 and OS. Since two piece modeling is used, PFS1, PFS2 and OS survival at the 24 month cutoff (per Methods model input) is needed

We agree. We have now revised our table 1 as requested based on the input by the esteemed reviewer. We have also included the statistical distribution and parameters with relevant information for the various survival curves for the model. 

We hope that our changes will meet the requirements of the journal and will help clarify any doubts the reviewer had and look forward to contributing to the journal soon.

Yours truly

Shaun Lee

On behalf of the authors

---

## [Decision Letter · Decision Letter 1]

21 Jan 2024

Cost-effectiveness analysis of olaparib maintenance therapy for BRCA mutation ovarian cancer in the public sector in Malaysia

PONE-D-23-29905R1

Dear Dr. Lee,

We’re pleased to inform you that your manuscript has been judged scientifically suitable for publication and will be formally accepted for publication once it meets all outstanding technical requirements.

Kind regards,

Alvaro Galli

Academic Editor

PLOS ONE

Additional Editor Comments (optional):

Reviewers' comments:

Reviewer's Responses to Questions

**Comments to the Author**

1. If the authors have adequately addressed your comments raised in a previous round of review and you feel that this manuscript is now acceptable for publication, you may indicate that here to bypass the “Comments to the Author” section, enter your conflict of interest statement in the “Confidential to Editor” section, and submit your "Accept" recommendation.

Reviewer #1: All comments have been addressed

2. Is the manuscript technically sound, and do the data support the conclusions?

Reviewer #1: Yes

3. Has the statistical analysis been performed appropriately and rigorously? 

Reviewer #1: Yes

4. Have the authors made all data underlying the findings in their manuscript fully available?

Reviewer #1: Yes

5. Is the manuscript presented in an intelligible fashion and written in standard English?

Reviewer #1: Yes

6. Review Comments to the Author

Reviewer #1: I am just writing some words here in order to meet your website's minimum character count. I don't actually have any comments to make.

7. PLOS authors have the option to publish the peer review history of their article (what does this mean?). If published, this will include your full peer review and any attached files.

Reviewer #1: No

---

## [Editor Report · Acceptance letter]

24 Jan 2024

PONE-D-23-29905R1 

PLOS ONE

Dear Dr. Lee, 

I'm pleased to inform you that your manuscript has been deemed suitable for publication in PLOS ONE. Congratulations! Your manuscript is now being handed over to our production team.

Kind regards, 

on behalf of

Dr. Alvaro Galli 

Academic Editor

PLOS ONE